# Next Viable Routes to Targeting Pancreatic Cancer Stemness: Learning from Clinical Setbacks

**DOI:** 10.3390/jcm8050702

**Published:** 2019-05-17

**Authors:** Kelvin K. Tsai, Tze-Sian Chan, Yuval Shaked

**Affiliations:** 1Laboratory of Advanced Molecular Therapeutics, Division of Gastroenterology, Department of Internal Medicine, Integrative Therapy Center for Gastroenterologic Cancers, Wan Fang Hospital, Taipei Medical University, Taipei 11696, Taiwan; fzesian@tmu.edu.tw; 2Graduate Institute of Clinical Medicine, School of Medicine, College of Medicine, Taipei Medical University, Taipei 11031, Taiwan; 3National Institute of Cancer Research, National Health Research Institutes, Miaoli 35053, Taiwan; 4Department of Cell Biology and Cancer Science, Rappaport Faculty of Medicine, Technion Integrated Cancer Center, Technion—Israel Institute of Technology, Haifa 3525433, Israel; yshaked@technion.ac.il

**Keywords:** cancer stemness, pancreatic ductal adenocarcinoma, stroma, therapeutics, clinical trials

## Abstract

Pancreatic ductal adenocarcinoma (PDAC) is a devastating and highly aggressive malignancy. Existing therapeutic strategies only provide a small survival benefit in patients with PDAC. Laboratory and clinical research have identified various populations of stem-cell-like cancer cells or cancer stem cells (CSCs) as the driving force of PDAC progression, treatment-resistance, and metastasis. Whilst a number of therapeutics aiming at inhibiting or killing CSCs have been developed over the past decade, a series of notable clinical trial setbacks have led to their deprioritization from the pipelines, triggering efforts to refine the current CSC model and exploit alternative therapeutic strategies. This review describes the current and the evolving models of pancreatic CSCs (panCSCs) and the potential factors that hamper the clinical development of panCSC-targeted therapies, emphasizing the heterogeneity, the plasticity, and the non-binary pattern of cancer stemness, as well as the desmoplastic stroma impeding drug penetration. We summarized novel and promising therapeutic strategies implicated by the works of our groups and others’ that may overcome these hurdles and have shown efficacies in preclinical models of PDAC, emphasizing the unique advantages of targeting the stroma-engendered panCSC-niches and metronomic chemotherapy. Finally, we proposed feasible clinical trial strategies and biomarkers that can guide the next-generation clinical trials.

## 1. Introduction

Pancreatic ductal adenocarcinoma (PDAC) is a highly lethal malignancy and currently the fourth leading cause of cancer death in the U.S. [1]. The majority of the patients with PDAC present with inoperable, advanced, and/or metastatic disease that are treatment-refractory. Recent advances in nanoparticle-formulated chemotherapy, such as albumin-bound paclitaxel and liposome-encapsulated irinotecan, and immunotherapy only provide limited survival benefit to a small fraction of the patients [2,3]. Even the majority of patients with operable disease develop recurrent and/or metastatic diseases within one to two years following surgery. Therefore, the developments of rational therapeutic strategies targeting the driver pathways of tumor aggressiveness and metastasis is critical for further improving the outcome of patients with PDAC. 

The stem cell model of tumorigenesis maintains that tumors are hierarchically organized and only a small population of cancer cells termed tumor-initiating cells (TICs) or cancer stem cells (CSCs) with a self-renewing ability of stem cells have the ability to initiate and sustain tumor growth [4,5]. A growing body of evidence accumulated over recent years, including large-scale genomic analysis and single cell RNA sequencing analysis, have consistently indicated the existence of stem-like cancer cells in a variety of both hematological malignancies and solid tumors. Among the solid tumors investigated, there are malignant glioma, breast cancer, prostate cancer, prostate cancer, non-small cell lung cancer, colorectal cancer (CRC), hepatocellular carcinoma, and PDAC [6,7,8,9,10]. Laboratory and animal studies have provided compelling data supporting CSCs as the driving force of cancer initiation, growth, metastasis, and treatment resistance [11,12,13]. Interestingly, CSCs were found to exhibit hallmarks of an epithelial-mesenchymal transition (EMT), a developmental program that enables cancer cell dissemination and seeding at metastatic sites [14,15,16]. Consistently, forced induction of the EMT programs in cancer cells imparts them with stem cell-like features, thereby promoting their metastatic and tumor-initiating properties [15,17,18]. An emerging paradigm entails a further level of hierarchy in CSCs with respect to their ability to proliferate or metastasize to distant organs [19,20]. The CSC heterogeneity can be exemplified by the alternative mesenchymal- and epithelial-like states of CSCs in breast cancer, which interestingly can transition between each other [19,21]. In human PDAC tissue, a distinct subpopulation of C-X-C motif chemokine receptor (CXCR)-4^+^ CSCs has been found to have a higher migratory potential and are capable of initiating liver metastasis compared with other CSC subpopulations [22]. Furthermore, PDAC progression was found to be driven by distinct sets of CSCs in temporally restricted bursts with little overlap between subsequent xenograft generations [23]. This heterogeneity and plasticity of CSCs make their therapeutic targeting highly challenging [24].

One of the hallmark features of PDAC is the pronounced stroma reaction termed “the desmoplastic response”, which comprises abundant cancer-associated fibroblasts (CAFs) and a highly dense fibrotic stroma [25]. CAFs are pro-inflammatory due to activation of nuclear factor kappa B (NF-κB), signal transducer and activator of transcription (STAT)-1 and STAT-3, and transforming growth factor (TGF)-β/SMAD signaling and are thereby engaged in active cross-talk with cancer cells through paracrine signaling involving chemokines, prostaglandins (PGE), insulin-like growth factor (IGF), and proteases [26,27,28,29,30,31], thereby promoting tumor angiogenesis, growth, and aggressiveness. Distinct subpopulations of CAFs are especially proficient in secreting a multitude of pro-stemness paracrine factors, including interleukin (IL)-6, IL-8, leukemia inhibitory factor (LIF), IGF-2, PGE-2, C-C motif chemokine ligand (CCL)-2, and nodal [27,32,33,34,35,36,37], thereby promoting the conversion of cancer cells into CSCs or supporting the self-renewal and the stemness properties of existing CSCs in tumors. Furthermore, systemic chemotherapy, such as gemcitabine therapy, can modulate CAFs in PDAC, which then acquire a senescence-like secretory phenotype to produce large amounts of pro-stemness chemokines to further enhance tumor stemness and aggressiveness following therapy [38]. Studies have demonstrated that CAFs are a heterogeneous population, with distinct function within tumors and metastasis. It has been shown that bone marrow derived CAFs specifically contribute to tumor angiogenesis unlike resident CAFs [39]. Aside from CAFs, the stroma of PDAC also comprise of bone-marrow derived mesenchymal stem cells (MSCs) recruited into the growing tumors [40]. Like CAFs, MSCs can significantly influence tumor behaviors and contribute to tumor progression. MSCs promote cancer stemness by secreting a specific panel of pro-stemness cytokines, chemokines, and growth factors or indirectly by differentiating into pro-stemness CAFs [41,42,43]. It is widely accepted that the desmoplastic stroma in PDAC constitutes a major obstacle for the efficient transport of cancer therapeutics into the tumor. Thus, therapeutics designed to directly target pancreatic CSCs (panCSCs) should be able to penetrate through the thick layer of fibrotic stroma to reach their target cells and exert their inhibitory effects. Otherwise, it may be more desirable to develop therapies directed at the stroma cells that help engender the niche microenvironments of panCSCs.

Not only CAFs and MSCs can engender pro-stemness niches in the stroma of PDAC, infiltrating immune cells, especially tumor-associated macrophages (TAMs), also form pro-panCSC niches by secreting a specific panel of pro-stemness factors [44,45,46,47]. Various small-molecule inhibitors of the Colony stimulating factor 1 receptor (CSF-1R) or C-C motif chemokine receptor (CCR)-2, which mediate the recruitment of TAMs to the tumor, or inhibitors of TAM-derived pro-stemness factors have shown anti-stemness and anti-tumor efficacies in preclinical studies [44,47,48]. Most encouragingly, a CCR-2 inhibitor has recently demonstrated activity in a phase 1b study [44], highlighting the clinical potential of TAM-targeted agents in the treatment of PDAC.

Developmental signaling pathways such as Wnt, Notch, TGF-β, and sonic hedgehog (SHH), are frequently co-opted by malignant cells during their transformation process into CSCs [49]. Wnt signaling plays an especially important role in the regulation of panCSCs [50,51]. Wnt signals constitute the important signals in the niche environments of CSCs in solid tumors, including PDAC [24,52]. Moreover, PDAC cells, especially panCSCs, develop a high responsiveness to Wnt signals due to aberrations in the Wnt signaling components [50,53,54,55,56]. Thus, the Wnt-related niche of panCSCs represents another potential viable therapeutic target in PDAC.

In this review, we describe the classical and the evolving models of CSCs in PDAC and their cell-intrinsic and extrinsic regulatory pathways. We highlight the unique drug-penetration barrier caused by the desmoplastic stroma of PDAC and the emerging insights into the heterogeneity, plasticity, and non-binary continuity of cancer stemness, which may together account for the setbacks in clinical trials testing therapeutics designed to directly target panCSCs. We propose potentially viable approaches to overcoming the current hurdles in targeting PDAC stemness, emphasizing stroma- and niche-targeting, nanotherapeutics, and metronomic chemotherapy. This review finally lists potential biomarkers that may optimize the clinical trial design and guide patient selection, which together may increase the success rate of developing anti-CSC therapies in PDAC.

## 2. CSCs in PDAC: An Evolving Model

The first exact proof of the existence of cancer stem cells (CSCs) was presented in 1994 by John Dick and colleagues, who successfully identified and purified human acute myeloid leukemia-initiating cells with distinct stem cell properties [57]. Based on the same concept, CSCs in solid tumors are functionally defined by their ability to self-renew, differentiate, and generate tumors recapitulating their parental ones [58]. However, due to the limitations of traditional xenotransplantation assays to characterize CSCs, highly sensitive and specific markers of CSCs are still lacking [24]. One popular method that has been widely used to enrich primary cancer cells for CSCs is culturing cells as anchorage-independent multicellular spheres or “tumorspheres” under serum-free culture conditions [22]. However, this functional assay does not allow for the detection of CSCs in vivo. A large body of studies have thus used cell-surface or intracellular antigens and proteins, many implicated from adult stem cell research, as markers to enrich for CSCs in various solid tumors, which also applies for the study of panCSCs. The early studies reported that PDAC cells that co-express the surface markers CD44 and CD133 or CD44, CD24 and epithelial-specific antigen (ESA) contain the enriched panCSCs [8,22,56]. In a mouse transgenic model of PDAC, a subset of CD133^+^CD44^high^ stem-like cells survived oncogene (Kras^G12D^) ablation and were responsible for tumor relapse [59]. In another transgenic model of pancreatic intraepithelial neoplasia (PanIN), the tuft cell marker calcium/calmodulin-dependent kinase-like (DCLK)-1 was found to mark a distinct population of TICs or panCSCs [60]. Aside from surface markers, panCSCs can also be enriched by their high aldehyde dehydrogenase 1 (ALDH-1) activity, and the presence of ALDH-1-positive panCSCs has been associated with poor prognosis in patients with PDAC [61]. Interestingly, ALDH-1 seems to define a population of panCSCs that are more tumorigenic than those defined by CD133, CD44, and/or CD24 [62]. Consistently, the presence of ALDH-positive tumor cells in the circulation has been associated with worse survival in patients with PDAC [63]. On the other hand, positive CXCR-1 expression has been correlated with lymph node metastasis and poor survival of patients with PDAC, which was attributed to the role of the IL-8/CXCR-1 axis in the regulation of panCSCs [64]. Of particular interest is the differential contributions of different CD44 isoforms to PDAC progression and stemness. It has been shown that the standard isoform of CD44 (CD44s) is associated with an EMT phenotype of PDAC cells, cancer invasiveness, and gemcitabine resistance [65]. This finding contrasts with another study reporting the expression of the CD44 variant 6 isoform (CD44v6) in metastatic PDAC and its role in EMT and metastasis in CRC [66,67]. This discrepancy may be explained by the differential pro-stemness roles of different CD44 isoforms in different types of cancers. Otherwise, it may reflect the inherent problems and the limitations of using surrogate markers, not pathway- or biology-informed ones, for the identification and the characterization of CSCs.

Other than the diversified markers used to define CSCs and panCSCs, recent insights into the characterization of CSCs argue against the traditional binary classification of cancer cells into stem versus non-stem cancer cells [68]. In fact, the CSC phenotype defined using current criteria may simply represent the extreme of a continuum of cellular phenotypic changes [11,12,13]. Even within the currently defined CSC populations, emerging data suggests that there is a further level of hierarchy with respect to the ability of CSCs to proliferate or to metastasize to distant organs [19,20,69]. For instance, in breast cancer, the subpopulation of CD44^+^CD24^−^ CSCs were found to be mesenchymal-like, non-proliferative, and pro-metastatic, whereas the non-overlapping subpopulation of ALDH^+^ CSCs are epithelial-like and highly proliferative [19]. In CRC, CSCs expressing the surface marker CD26 was responsible for liver metastasis [70]. Echoing this emerging paradigm of CSC heterogeneity, the previously identified panCSCs populations have been shown to contain largely non-overlapping CSC subpopulations. For instance, an analysis on freshly isolated human PDAC cells revealed that less than 1% of CD44^+^CD24^+^ panCSCs overlap with those that are ALDH^+^ [61]. Similarly, there is minimal overlap between DCLK-1^+^ panCSCs and ALDH^+^ cells in PanIN [60]. A distinct subpopulation of CXCR-4^+^ panCSCs localized to the invasive front of human PDAC tissues have a high migratory potential and are capable of initiating liver metastasis [22]. Other studies demonstrated that cells that contain a side population as assessed by flow cytometry, are considered CSCs due to their characteristic in efflux cytotoxic drugs and thus contribute to resistance to chemotherapy [71]. As such, a side population of PDAC cells has been shown to resist gemcitabine therapy and express stemness-associated and prognostic genes [72,73]. Aside from being highly heterogeneous, the CSC phenotype has been shown to be highly dynamic and plastic such that different CSC populations can convert into each other [74]. For instance, a considerable proportion of the mesenchymal-like and epithelial-like subpopulations of CSCs in breast cancer can transition between each other [19,20,21]. Using a serial xenotransplantation assay, long-term progression of PDAC was found to be mediated by distinct sets of CSCs in temporally restricted bursts with little overlap between subsequent xenograft generations [23]. Aside from the interconversion between CSC subpopulations, CSCs can also be transdifferentiated from differentiated cancer cells through cellular reprogramming [34], which may be facilitated by cytotoxic stresses such as chemotherapy, ionizing radiation, and the genetic ablation of pre-existing CSCs [38,75,76,77]. These latest findings reshape our understanding of the regulation of CSCs and implicate that identifying different panCSC subsets and elucidating their roles in malignant progression will not only improve our understanding of how cancer stemness is fine-tuned in PDAC but also may disclose novel targets based on which panCSC-targeted therapeutic strategies can be developed and deployed in the clinic. We thus envisage that more rigorous approaches to define and characterize panCSCs, such as the lineage-tracing method used in the study of CSCs in CRC and cutting edge single-cell sequencing or “omics” methodologies used in the study of glioma stem cells, should be applied to more unambiguously define panCSCs and their functional roles in PDAC biology and progression [13,24].

## 3. The Molecular Pathways Controlling PanCSCs

CSCs, including panCSCs, are coordinately regulated by a complex network of cell-intrinsic and extrinsic factors. The recent insights into the regulation of CSCs revealed that they exist in a dynamic equilibrium with cells and factors within different “niche” microenvironments, such as the hypoxic niche, the perivascular niche, the immune niche, and the tumor invasive front [68]. The best studied cell-extrinsic factors regulating CSCs are inflammatory cytokines and chemokines, such as IL-6, IL-8, LIF, and CCL-5, which have been shown to play an essential role in CSC regulation and the invasion and metastasis of tumors [32,33,34,37]. Although the niche microenvironments for panCSCs are less well defined than those in other types of cancers, two types of stroma cells, including pancreatic stellate cells (PSCs), a specialized type of CAFs in PDAC, and their precursor MSCs have drawn particular attention given that they are the major cellular components of the desmoplastic stroma in PDAC [40,78,79]. For instance, PSC-derived IL-8 has been shown to profoundly enhance the stemness property of PDAC cells [62,64,80]. PSCs also secrete the transforming growth factor (TGF)-β family protein nodal, which binds to its receptors Activin-like (Alk)-4 and Alk-7 on panCSCs to promote their stemness properties [36,81]. A recent proteomic screening identified LIF, rather than IL-6 or other interleukins, as the major pro-panCSC factors secreted by PSCs. Specifically, PSC-derived LIF induced STAT-3 signaling in PDAC cells, leading to activation of stemness programs [37]. Notably, following systemic chemotherapy, PSCs secrete large amounts of the ELR^+^ CXCL chemokines through chronic activation of the STAT-1 and NF-κB transcriptional activities, which stimulated CXCR-2 signaling in cancer cells to elicit their transdifferentiation into CSCs and thereby promoted post-treatment tumor aggression and treatment failure [38]. Similarly, in a mouse model of PDAC, the number of bone marrow-derived MSCs significantly increased following gemcitabine treatment in the tumor stroma, and the gemcitabine-educated MSCs have a positive regulatory effect on CSCs through the STAT-3–CXCL-10–CXCR-3 paracrine signaling axis [82].

Recently, paracrine factors derived from TAMs were also found to have pro-stemness functions in PDAC. For instance, TAMs in the stroma of PDAC secrete oncostatin M, an IL-6 family protein [46], which potentiates the EMT and the stemness programs in PDAC cells by inducing the expression of the EMT regulators zinc finger E-box binding homeobox (ZEB)-1 and snail [83]. TAMs in PDAC also secrete the immune-modulatory peptide leucine leucin-37 (LL-37) and the interferon-stimulated factor ISG-15, which act on panCSCs to promote their tumorigenicity and invasiveness [45,47].

The cell-intrinsic pathways controlling the CSC phenotype are frequently development signaling pathways, including notably TGF-β, Wnt, SHH, and Notch signaling [16]. Among them, Wnt signaling plays an especially important role in the regulation of panCSCs and has been strongly implicated in the progression and the metastatic colonization of PDAC [50,51]. Metastatic PDAC cells in ascites and the blood circulation, which contain the enriched panCSCs, express high levels of Wnt signaling genes, such as Wnt-2 [50]. Another study identified the novel oncoprotein family with sequence similarity 83 member A (FAM-83A), which promotes PDAC stemness and chemoresistance by activating Wnt/β-catenin and TGF-β signaling [84]. A high Wnt/β-catenin transcriptional activity has also been linked to lympho-vascular invasion in human PDAC [54]. Mechanistically, Wnt signaling is activated in PDAC through multiple genetic lesions in the pathway, ranging from the ligand (e.g., WNT1, WNT2, WNT5A, WNT7A), the receptor (e.g., FZD2, FZD7), and/or the effector levels (e.g., CTNNB1, APC, AXIN1) [50,85,86,87]. Wnt signaling can be also activated in PDAC through protein regulation, such as that mediated by ATDC (ataxia-telangiectasia group D complementing) or ASPM (abnormal spindle-like microcephaly associated) [56,88]. Specifically, ASPM augments canonical Wnt–β-catenin signaling and PDAC stemness through stabilizing dishevelled (Dvl)-2, an upstream hub in Wnt signaling [56].

A few studies have implicated the roles of Notch and SHH pathways in the regulation of panCSCs. PanCSCs have been shown to express high levels of the Notch pathway components, including Notch-1, Notch-2, Notch-3, Jagged 1, Jagged 2, and Delta like canonical Notch ligand (DLL)-1; therefore, inhibiting Notch signaling by gamma secretase inhibitors reduced the percentage of panCSCs and their stemness property [89], leading to a strong anti-tumor efficacy in PDAC [90]. Conversely, the same study also showed that stimulation of Notch pathway increased the percentage of panCSCs. In PanIN, Notch signaling regulates DCLK-1 expression, which exerts a tubulin-acetylation activity indispensable for the clonogenic potential of panCSCs [60]. On the other hand, SHH signaling has been shown to specifically regulate ALDH^+^ panCSCs and PDAC metastasis [91]. Another study showed that SHH signaling components, including smoothened and GLI family zinc finger 1 (GLI-1), are required for hypoxia-induced EMT in PDAC cells [92]. Aside from developmental pathways, several novel regulators of panCSCs have emerged recently. First, the liver-specific transcriptional factor, HNF1 homeobox A (HNF1A), has been found to regulate panCSCs through directly controlling the expression of the pluripotency factor Octamer-binding transcriptional factor (OCT)-4 [93]. Second, genetic and PDX models of PDAC have led to the identification of Musashi (Msi), a RNA binding protein and stem cell regulator, as a critical regulator of panCSCs and PDAC progression through controlling the expression of a panel of stem cell regulators, proto-oncogenes, and regenerating family genes, such as *MET*, *BRD4*, and *HMGA2* [94]. A subsequent genomic and CRISPR screening based on Msi^+^ panCSCs identified the nuclear hormone receptor retinoic-acid-receptor-related orphan receptor (ROR)-γ as a regulator of panCSCs, and its pharmacologic blockade reduced the number of panCSCs and their tumorigenic potential and inhibited the growth of PDAC [95].

Epigenetic mechanisms, especially microRNAs (miRNAs), may also play important roles in controlling panCSCs by regulating stemness pathways. For instance, miRNA-1181 suppressed panCSCs by targeting the pluripotency factor SRY-box (SOX)-2 and STAT-3 [96]. Alternatively, a genomic screening has identified miRNA-21 and miRNA-221 as upregulated miRNAs in panCSCs and their targeting using antisense oligonucleotides reduced the percentage of panCSCs along with the invasion and the chemoresistance of PDAC cells [97]. Likewise, miRNA-1246 was found to be up-regulated in panCSCs and contribute to their tumor-initiating potential and the induction of drug resistance [98]. Conversely, miRNA-17-92 was found to be a downregulated miRNA cluster in panCSCs; therefore, its overexpression reduced the self-renewal capacity of panCSCs and reversed tumorigenicity and chemoresistance by targeting Nodal/Activin/TGF-β signaling [99]. Another under-expressed miRNA in panCSCs is miRNA-335, which targets the pluripotency regulator OCT-4. Accordingly, the systemic delivery of miRNA-335 inhibited PDAC metastasis [100].

## 4. The Major Hurdles in the Therapeutic Targeting of Pancreatic Cancer Stemness

Although a number of druggable targets of CSCs have been identified and many CSC-directed therapies have been developed [101], the field and the industry have witnessed a series of clinical trial setbacks and failures over the past decade. Notable failures included the focal adhesion kinase (FAK) inhibitor defactinib, the STAT-3 inhibitor napabucasion, the anti-Notch-2/3 antibody tarextumab, the anti-DLL-4 antibody demcizumab, and most recently the anti-DLL-3 antibody-drug conjugate rovalpituzumab tesirine (Rova-T). Moreover, the clinical trial combing the SHH pathway inhibitor saridegib, another potential CSC regulator, and gemcitabine in the treatment of PDAC was discontinued due to the worse survival of patients treated with the combination therapy than those treated with chemotherapy alone. It is thus imperative to investigate into the potential mechanistic explanations underlying these clinical setbacks before the continuing development and the clinical studies of next-generation anti-CSC therapies.

Regarding the therapeutic targeting of panCSCs and PDAC stemness, several specific issues should be taken into consideration (Figure 1). First, as discussed above, CSCs, including panCSCs are highly heterogeneous and phenotypically plastic and their different subpopulations can interconvert into each other [74]. Most importantly, CSCs can be directly converted from differentiated cancer cells through transdifferentiation, which can be especially triggered by cytotoxic therapy [34,38,75]. The highly dynamic characteristics of CSCs make them moving targets in anti-cancer therapy, presenting a daunting challenge to therapeutic efforts aiming at eradiating them. Indeed, two elegant studies have reported that ablation of CSCs only temporarily halted tumor growth, whereas the tumors could resume growth following the removal of the cell death inducers due to the re-emergence of CSCs from differentiated tumor cells [76,77]. These results call into question whether the direct targeting of CSCs remain a viable option in cancer treatment. Second, the florid desmoplastic reaction in the stroma of PDAC represents a formidable barrier to any therapeutics designed to target the small or rare population of panCSCs spaced within tumor nests [25]. As such, therapeutics, especially antibodies or their derivatives, have very limited penetration into the desmoplastic stroma of PDAC; therefore, they may only reach the subsets of panCSCs spaced at the outer rim of tumors or those located near blood vessels. If so, their anticipated anti-CSC and anti-tumor effects will be severely crippled. Indeed, clinical data has confirmed that chemotherapy agents, such as gemcitabine, can only reach the stroma but not the tumor cells in human PDAC tissues [102]. Finally, since panCSCs only comprise a small or even a rare fraction of tumor cells, panCSC-directed therapies would not be expected to produce measurable changes in tumor burden according to conventional treatment response criteria. Thus, there is a pressing need for developing “stemness-informed” surrogate markers of response in order to guide the better assessment of clinical trials testing anti-CSC drugs, especially at the phase II stage [103]. With these challenges in mind, we provided here some insightful perspectives on potentially viable routes to effectively and safely targeting PDAC stemness (Figure 2).

## 5. Route 1: Targeting the Wnt-Related Niche of PanCSCs

Developmental pathways regulating self-renewal mechanisms during normal stem cell development, such as Notch, TGF-β, and SHH, are frequently co-opted by malignant cells during their malignant transformation process and the acquisition of the stemness phenotypes [49]. During the past two decades, therapeutics targeting these pathways have been developed and tested in clinical trials. Unfortunately, as described above, most of these agents yielded only marginal or disappointing anti-tumor efficacy in clinical studies. These setbacks raised the possibility that therapeutics targeting developmental pathways, which also play critical roles in normal somatic stem cell homeostasis, are associated with a relatively small therapeutic index, casting doubt on their clinical feasibility and validity.

One viable option of targeting developmental pathways is the targeting of the niche microenvironments they help foster. Of particular relevance to this direction is the targeting of Wnt-related CSC niche as Wnt factors have been reported to constitute the important signals in the niche environments of CSCs in CRC and PDAC [24,52]. Reinforcing this unique therapeutic opportunity, a considerable subset of PDAC cells, especially panCSCs, are known to have a heightened Wnt responsiveness compared with normal cells or other non-stem cancer cells, which may further increase the therapeutic index of Wnt-targeted agents. For instance, a subset of PDAC carries inactivating mutations of the negative regulator of Wnt signaling, Ring finger 43 (RNF-43), which is an ubiquitin E3 ligase mediating the degradation the Frizzled receptors [50,53]. Other PDAC upregulate the expression of Wnt-7B, Wnt-2, or the novel Wnt regulator ATDC, which positively regulates Dvl-2 to mediate cell-autonomous Wnt/β-catenin activation [54]. Along this line, our group identified a novel Wnt co-regulator ASPM, whose expression is up-regulated in panCSCs. Specifically, ASPM renders panCSCs highly responsive to Wnt signals by positively regulating Dvl-2 and β-catenin [55,56].

During the past two decades, a number of Wnt targeted agents, including small-molecules and antibody therapeutics, have been developed with a couple of them entering clinical trials [104,105,106]. Some of the Wnt inhibitors had demonstrated anti-CSC efficacy, including the antagonist of the β-catenin–transcription factor (TCF) interaction and the interaction between β-catenin and its coactivator CREB binding protein (CBP) [104,107]. However, the direct targeting of CSCs by antagonizing Wnt signaling remains highly challenging, which is exemplified by the disappointing clinical development of the anti-Frizzled antibody vantictumab and the Frizzeld-8–Fc fusion protein ipafricept, which had recently been discontinued from the pipeline. According to a phase 1b study in triple-negative breast cancer, vantictumab is well tolerated (NCT01973309); therefore, the lack of clinical efficacy might have led to its deprioritization.

Targeting Wnt-related niches may provide an attractive and more theoretically feasible approach especially in desmoplastic cancer such as PDAC as the tumor stroma is more accessible to therapeutic agents diffused from the blood circulation [108]. In this regard, Wnt-related niches have been identified in several types of cancers. For instance, in a transgenic model of colon adenoma, the growth of Leucine rich repeat containing G protein-coupled receptor (LGR)-5^+^ stem cells was found to depend on Wnt-3 derived from their niche cell Paneth cells, which could be inhibited by treatment with a Porcupine inhibitor [109]. Porcupine is a membrane-bound O-acyltransferase mediating the palmitoylation of Wnt ligands essential for their secretion. Several Porcupine inhibitors, including LGK974 (Novartis), RXC004 (Redx Pharma), and ETC-1922159 (A*STAR, Singapore), has shown preclinical efficacy in breast cancer, head-and-neck squamous cell carcinoma and CRC, are now under phase I development [110,111]. Notably, in PDAC, LGK974 was found to specifically inhibit the tumor cells carrying inactivating mutations of RNF-43, which activates Wnt signaling in these cells [53]. This finding raised the possibility that Porcupine inhibitors may exert their efficacy in cells with heightened Wnt activity, including panCSCs as they have a heightened responsiveness to Wnt signals [50,56,85,86,87], which merits further investigations in preclinical and clinical studies (Table 1).

Aside from small-molecule inhibitors, nanoparticle-dependent delivery of therapeutics may serve as another valid approach to target the Wnt-related niches of panCSCs. Nanoparticles can facilitate the delivery of therapeutics into or through the desmoplastic stroma of PDAC. The cargo that can be delivered by nanoparticles include compounds, peptides or oligonucleotides, such as small interfering RNA (siRNA), microRNA, synthetic antisense oligonucleotide (ASO), and plasmid DNA, that is designed to antagonize Wnt factors present in the niche microenvironment or affect their production from niche cells. The clinical feasibility and the theoretical advantages of nanotherapy in PDAC will be discussed in details in the following section.

## 6. Route 2: Targeting Pro-Stemness PSCs

CAFs in desmoplastic cancers, such as PSCs, are proficient in paracrine signaling and are capable of secreting a multitude of paracrine factors that maintain and expand CSCs. The pro-stemness factors released by CAFs and PSCs include IL-8 [62,64], which regulates a subpopulation of epithelial-like CSCs that express high ALDH activity [112], the TGF-β family protein Nodal/Activin-A, which binds to its receptor Alk-4 and Alk-7 on CSCs to promote their stemness properties [36,81], IL-6, CXCL-1, and CXCL-2 [113], which regulates the stemness phenotype through the STAT-3–NF-κB signaling pathway [33,34,114]. Recent studies pointed to the functional and phenotypic heterogeneity in PSCs. First, two distinct subgroups of PSCs have been identified in human PDAC tissues [115]. Only those PSCs located away from tumor cells, denoted as “inflammatory CAFs (iCAFs)”, were proficient in secreting pro-stemness IL-6, CXCL-1 and CXCL-2 through activation of IL-1α–Janus kina (JAK)–STAT signaling [113]. Second, our group further uncovered that, following systemic chemotherapy, the residual PSCs or CAFs were “locked” into a senescence-like and chronic inflammatory state, secreting large amounts of pro-stemness ELR^+^ CXCL chemokines, including CXCL-1, CXCL-2, CXCL-5, and CXCL-6. Importantly, these therapy-modulated PSCs triggered the transdifferentiation of PDAC cells into panCSCs and thereby promoted tumor aggression and treatment failure [38].

In analogous to the targeting of Wnt-related panCSC niche, targeting the pro-stemness PSCs have multiple theoretical advantages over the direct targeting of panCSCs. First, as stressed repeatedly in this review, CSCs are highly heterogeneous and plastic [34,38,74,75], making them moving targets and hard to be completely eradicated. By contrast, CAFs are both genetically and phenotypically stable; therefore, CAF-directed therapies may have more stable and sustainable anti-stemness effects. Second, the recent discoveries of the specific subpopulations of pro-stemness CAFs and PSCs not only provides novel therapeutic targets, such as GPR-77 [116], but also may render the CAF-targeted therapies more specific and safe than the overall elimination of CAFs, which had led to paradoxical tumor invasion and immunosuppression [117,118]. Third, PSCs comprise a large fraction of the PDAC stroma, which contrasts sharply with CSCs that comprise only a small or even a rare subpopulation of cancer cells that exist within isolated cancer cell nests or as individual dispersed cells [16,24]. Indeed, it has been reported that PSCs may account for more than 90% of the total tumor volume of PDAC [78,119]. Thus, at a given tissue concentration, there will be a far larger number of PSCs that are exposed to therapeutics than that of CSCs. In addition, the spatial distribution of PSCs also provides another advantage in their therapeutic targeting. PSCs are often localized to the periphery of the tumor bed or glands and close to blood vessels, rendering them directly accessible to the therapeutics diffused from the blood circulation [108]. The positive regulatory role of PSCs in PDAC stemness, together with their ideal “strategic deployment” in the tumor stroma and their genetic stability, make PSC-directed anti-stemness therapy potentially more viable than the direct panCSC targeting.

Previously, diversified approaches have been exploited to block the PSC–panCSC communication or the downstream signaling events in panCSCs. For instance, systemic administration of a LIF-neutralizing antibody in combination with chemotherapy has been recently shown to reduce the percentage of CSCs and extended the survival of tumor-bearing mice in a transgenic model of PDAC [37]. A high-affinity anti-IL-6 antibody, MEDI5117, has been shown to enhance the anti-tumor efficacy of chemotherapy in several types of tumors that are known to be driven by the IL-6–STAT-3 signaling [120]. The small-molecule inhibitors of CXCR-2, the receptor of IL-8 and the ELR^+^ CXCL chemokines, including AZ13381758 and SB225002, have shown preclinical efficacy in transgenic or PDX models of PDAC [38,121]. Furthermore, a small-molecule inhibitor of STAT-3, BBI608, has been reported to significantly inhibit cancer stemness in a variety of cancer types, including PDAC [122]. It should be noted that, as aforementioned, large-molecule therapeutics, such as the anti-LIF and anti-IL-6 antibodies, may have very limited penetration into desmoplastic tissues and may only be able to reach CAFs spaced at the outer rim of tumors or those located surrounding or near blood vessels. The direct targeting of panCSCs by such as the CXCR-2 and the STAT-3 inhibitors is challenging due to their highly dynamic and plastic characteristics [74]. As such, compared with the above-mentioned approaches, small-molecule inhibitors or nanoparticle-formulated therapeutics designed to target the pro-stemness PSCs may be more viable and feasible approaches. In line with this possibility, the small-molecule TGF-β inhibitor SD208 has been shown to reduce the CAF-induced expression of stemness markers while induced the expression of differentiation markers in CAF-cocultivated CRC cells. Accordingly, SD208 in combination with the small molecule inhibitor of the SHH pathway transcriptional factor GLI-2 could restore the sensitivity of the tumors to chemotherapy [123]. In another example, Vitamin D receptor (VDR) signaling has been shown to antagonize the TGF-β/SMAD signaling-induced activation of PSCs mediated by IL-6, CCL-2 and CXCL-1 in PDAC [31]. Calcipotriol, a potent vitamin D analog that controls VDR induction, inhibited inflammatory signaling in PSCs and reduced their expression of IL-6, CCL-2, and CXCL-1. As such, calcipotriol could synergize with chemotherapy to control tumor growth and extend survival in transgenic mouse models of PDAC. 

An alternative to the inhibition of PSC- or CAF-derived pro-stemness factors is their direct depletion, which had been achieved by an oral DNA vaccine targeting the CAF-specific marker fibroblast activation protein (FAP). This gene therapy approach has been shown to suppress tumor growth and metastasis in murine models of CRC and breast cancer [124]. Another approach involved the adoptive transfer of FAP-targeted chimeric antigen receptor (CAR) T cells, which killed FAP^+^ CAFs and induced multiple beneficial stroma alterations, leading to delayed tumor growth and survival extension in mouse models of NSCLC and PDAC [125,126]. Notably, the same study demonstrated that a combined targeting of FAP^+^ CAFs and EPH receptor A2 (EphA2)^+^ cancer cells led to a nearly complete remission of the tumors [125], suggesting that CAF or PSC targeting can synergize with cancer-cell killing. Notwithstanding these promising findings, it is worthy of noting that a complete eradication of PSCs from tumors has led to invasive and undifferentiated tumors along with unfavorable immunosuppression [117,118]. In light of this risk, functional inhibition of CAFs or the targeted depletion of their pro-stemness subset may be a safer and more desirable approach than the nonspecific depletion of all the CAFs or PSCs. Given the multiple advantages of PSCs or CAFs targeting and the continuing development of their targeting agents, we anticipate that more small-molecule drugs or novel therapeutics designed to target PSCs and the pro-stemness niches they engender will enter clinical trials in the near future. Since these stroma-targeted agents have better access to their target cells, they may be associated with a higher therapeutic index compared with agents designed to directly target panCSCs, thereby increasing the successful rate in early phase studies.

## 7. Route 3: Targeting Cancer Stemness by Nanotherapeutics

Since the desmoplastic stroma of PDAC constitutes a major obstacle for the efficient transport of cancer therapeutics into the tumor [25], a large proportion of intravenously administered therapeutics, including chemotherapeutic agent like gemcitabine, fail to reach and affect tumor cells. Recently, two nanoparticle-formulated chemotherapy agents, including albumin-bound paclitaxel (*nab*-paclitaxel; Abraxane^®^) and liposome-encapsulated irinotecan (Onivyde^®^), have shown anti-tumor efficacy and provided survival benefit to patients and was thus approved for the treatment of advanced PDAC [127,128]. Notably and importantly, both reagents could significantly increase the levels of the chemotherapeutic agents (i.e., paclitaxel and irinotecan, respectively) in the treated tumors [128,129]. These precious and successful clinical experiences strongly suggest that nanoparticle formulation is a clinically valid approach to improve the drug penetration and thus increase the treatment efficacy of desmoplastic cancers such as PDAC. Yet, not all encapsulated chemotherapies are considered effective drugs against PDAC. For example, liposomal doxorubicin (lipodox^®^) failed to demonstrate therapeutic activity in phase II studies in advanced PDAC patients (NCT00426127, NCT00609765) [130]. Aside from chemotherapy agents and compounds, nanoparticles may also facilitate the delivery of oligonucleotides, such as small interfering RNA (siRNA), microRNA, and DNA plasmid, into tissues. Nanoparticle-formulated oligonucleotide therapy has recently become a clinical reality as the liposome-encapsulated siRNA specific for transthyretin (Patisiran^®^, Alnylam Pharmaceuticals, Cambridge, MA, USA) has demonstrated a remarkable clinical efficacy in patients with liver-associated hereditary amyloidosis and thus became the first clinically approved nanoparticle oligonucleotide drug [131].

Several recent preclinical works have supported the feasibility and the validity of using nanotherapeutics to target the panCSC niches and the pro-stemness stroma in PDAC. In the first study, our group has demonstrated that bone marrow-derived MSCs reside in close proximity to panCSCs following gemcitabine chemotherapy and support the panCSC niche. Mechanistically, the gemcitabine-exposed MSCs secreted high levels of CXCL-10, which acted on its receptor CXCR-3 on panCSCs, activating STAT-3 signaling and promoting their survival. Importantly, systemic administration of the CXCL-10 inhibitor AMG487 formulated with novel MSC-derived membrane-based nanoparticles termed “nanoghost” led to the intratumoral accumulation of AMG487 in close proximity to panCSCs, thereby reducing the percentage of CSCs and augmenting the therapeutic efficacy of gemcitabine [82]. In the second study, systemic administration of a nanocarrier-formulated plasmid encoding a secretable form of the death ligand TNF-related apoptosis-inducing ligand (TRAIL) termed sTRAIL transduced PSCs and converted them into sTRAIL-producing cells. This strategy successfully triggered apoptosis of neighboring cancer cells and demonstrated strong anti-tumor efficacy in a PDAC model [132]. Presumably, similar strategies can be adopted to facilitate the intra-tumoral and the intra-stromal delivery of panCSC- or PSC-specific siRNAs or miRNAs or their inhibitory oligonucleotides as described above [96,97,98,99,100]. In the future, investigators in both the academic and the industrial sectors should seek the opportunity of formulating lead therapeutics with nanocarriers to facilitate their penetration into the stroma and the panCSC niches in PDAC, which we envisage will make a big leap forward for successfully targeting panCSCs and their niches.

## 8. Route 4: Targeting the Crosstalk between TAMs and PanCSCs

TAMs, especially their M2-like subset, are one of the major cellular components in the stroma of PDAC and have important pathogenetic significance. Indeed, clinical correlative studies have shown that the density of CD68^+^ or CD204^+^ TAMs or CD163^+^ M2-polarized TAMs correlated with lymph node metastasis or poor survival in patients with PDAC [133,134,135]. Consistently, a high ratio of CD68^+^ TAMs to CD8^+^ T cells correlated with poor survival in patients with PDAC [44]. Similarly, the ratio of peripheral blood to bone-marrow-derived inflammatory monocytes predicts decreased survival in patients with resected PDAC [48]. Mechanistically, TAMs exert pleiotropic effects on tumor cells by fostering an immune-suppressive microenvironment and activating PSCs in PDAC, thereby inhibiting the response of tumors to immunotherapeutic agents and promoting metastatic tumor growth [136,137]. Notably, systemic chemotherapy, such as gemcitabine, doxorubicin, and paclitaxel, further enhances the tumoral infiltration of TAMs, which has been shown to significantly contribute to the treatment resistance in PDAC and breast cancer [44,138,139]. Accordingly, blockage of TAM infiltration could synergize with chemotherapy to improve the response to immune-checkpoint inhibitors in PDAC [44,136].

Several recent studies have highlighted the important roles of TAMs in the regulation of PDAC stemness and panCSCs. For instance, coculture of M2-polarized TAMs with PDAC cells increased the frequency of ALDH^+^ panCSCs and the expression of stemness genes [44,45]. A clinical study has reported the positive correlation between the expression of CD44^+^CD133^+^ panCSCs and that of CD204^+^ TAMs in patients with PDAC [135]. As such, inhibiting the tumoral infiltration of TAMs by small-molecule inhibitors of CSF-1R (PLX6134 or PLX3397) or CCR-2 (PF04136309) could profoundly reduce the number of panCSCs, thereby improving treatment response and reducing peritoneal and hepatic metastasis in mouse models of PDAC [44,48]. Accordingly, several CSF-1R or CCR-2 inhibitors, including ARRAY382 (Array Biopharma) and PLX3397 (pexidartinib; Plexxikon), are in clinical development in solid tumors, including PDAC (Table 1). Interestingly, panCSCs were found to potently inhibit the proliferation of CD8^+^ T cells; therefore, inhibiting TAMs may not only block their crosstalk with panCSCs but also indirectly de-repress the immunosuppressive microenvironment in PDAC. Actually, the TAM–CSC communication is not unidirectional and there is reciprocal crosstalk between TAMs and panCSCs. Along this line, panCSCs were found to promote the polarization of TAMs toward an M2 phenotype, which in turn promoted the stemness of panCSCs and their tumorigenic potential. Similarly, a screening of the upregulated genes in the panCSC-educated TAMs has identified interferon-stimulated gene (ISG)-15, which could in turn reinforce the self-renewal, invasive, and tumorigenic potential of panCSCs [45]. Another study identified human cationic antimicrobial protein (hCAP)-18 and its cleavage product LL-37 as pro-stemness factors secreted by TAMs in PDAC [47]. TAMs secrete hCAP-18/LL-37 in response to panCSC-derived Nodal/Activin A and TGF-β1. The receptor of hCAP-18/LL-37, formyl peptide receptor (FPR)-2 and P2X purinoceptor 7 receptor (P2X7R), were found to be predominantly expressed on CD133^+^ panCSCs, suggesting that the TAM–panCSC communication through the LL-37 paracrine signaling is panCSCs-specific. As such, in a transgenic model of PDAC, blocking FPR-2 or P2X7R with their respective inhibitors WRW4 and KN62 could reduce the numbers of circulating tumor cells and inhibit tumorigenesis and liver metastasis [47]. Blocking the panCSC-derived Nodal/Activin A signaling with SB421542 and SB505124 may also cancel the induction of LL-37 expression on TAMs and thereby block the TAM–panCSC paracrine loop. Since the expression of hCAP-18/LL-37 was up-regulated in human PDAC tissues, its molecular targeting presents a promising and feasible opportunity of disrupting the TAM-related panCSCs niche. 

Recently, the clinical value of TAM targeting in the treatment of PDAC is credentialed by the encouraging result of a phase 1b trial of the CCR-2 inhibitor PF04136309 (Pfizer) [44]. Of the 33 patients with locally advanced PDAC who were treated with the standard chemotherapy regimen FOLFIRINOX and PF04136309, up to 32 (97%) of whom achieved local tumor control tumors and one patient presenting objective tumor response (Table 1). This promising result is worthy of further investigation and may encourage clinical studies of other TAM-targeted therapies, including those specifically involved in the TAM–panCSC crosstalk, in the treatment of PDAC. It should be noted that TAMs’ elimination in tumors have been a major challenge and treatment focus for the last decade due to its multi pro-tumorigenic biological processes within tumors, especially in response to therapy [140]. Therefore, clinical studies using macrophage inhibition are currently designed and executed.

## 9. Route 5: Metronomic Chemotherapy to Temper Therapy-Induced Cancer Stemness

Traditional protocols of chemotherapy involve administration of drugs to patients in single dose or short courses at their maximum tolerated doses (MTD). However, emerging laboratory and clinical evidences have unveiled the unique advantage of using comparatively low doses of chemotherapy drugs on a more frequent or continuous schedule, a concept commonly referred to as “low dose metronomic (LDM)” therapy [141,142,143]. Accumulating clinical evidence has supported the use of LDM therapy as an alternative for primary or maintenance chemotherapy as it offers an equal or even better anti-tumor efficacy than the traditional MTD regimens [141]. Mechanistically, the efficacy of LDM therapy has been originally and initially attributed to its anti-angiogenic effects [143,144], reduced recruitment of endothelial progenitors [145], and increased expression of thrombospondin-1 [146]. Recent evidence also suggests that LDM therapy mediates its anti-tumor effect by inhibiting regulatory T cells [147,148], by triggering the maturation of tumor-infiltrating dendritic cells [149], or by disrupting the vascular niches supporting CSCs [150].

There are now compelling evidences demonstrating that MTD chemotherapy also induces alterations in CAFs [138,151], and LDM chemotherapy is able to temper the therapy-induced stromal alterations in desmoplastic cancers such as PDAC and breast cancer [38,152,153]. In keeping with this paradigm, our group recently demonstrated the systemic MTD chemotherapy using assorted agents, including paclitaxel, gemcitabine, doxorubicin, and cyclophosphamide, had profound impacts on CAFs in human breast cancer and PSCs in PDAC. The chemotherapy-modulated CAFs/PSCs acquired a senescence-like phenotype and the ability to secret large amounts of pro-stemness ELR^+^ CXCL chemokines through chronic activation of STAT-1 and NF-κB signaling [38]. Importantly, the pro-stemness niche microenvironment generated by therapy-modulated CAFs/PSCs could be attenuated by pretreating the tumors with a CXCR-2 inhibitor or by switching the dosing schedule to LDM regimens. We envisage that the LDM therapy approach has multiple benefits in the development of anti-CSC therapies. First, it involves the use of standard and clinically approved chemotherapeutic agents and obviates the lengthy and costly process of developing new CAF- and/or CSC-targeted agents. Second, the advent of an increasing number of oral chemotherapeutic agents makes the concept of LDM chemotherapy immediately clinical applicable. Indeed, the clinical benefit of LDM chemotherapy has quickly accumulated over recent years. In breast cancer, LDM therapy has yielded an average response rate of 39% and an average overall clinical benefit of 57%. In CRC, a large and randomized phase III trial (CAIRO3 trial) provided a solid support for the clinical benefits of the maintenance use of capecitabine, an oral form of 5-fluorouracil (5-FU) [154]. In a randomized phase III study, another oral 5-FU drug S-1^®^ (Taiho Pharmaceutical, Tokyo, Japan) proved to be non-inferior to infusional gemcitabine in patients with locally advanced or metastatic PDAC [155], which led to its approval for the treatment of metastatic PDAC in Japan and Taiwan (Table 1). Subsequently, in large-scale phase III adjuvant trials, S-1^®^ and capecitabine both proved to be superior to infusional gemcitabine in prolonging the survival of patients with resected PDAC [156,157]. Third, as mentioned earlier, LDM chemotherapy not only may prevent the dangerous duet of CAFs and CSCs but may also exert multiple favorable effects on other cells in the tumor stroma, including TAMs, myeloid-derived suppressor cells (MDSCs) and blood vessel cells [152,153,158,159]. Taken together, we foresee that LDM chemotherapy will become the treatment of choice in many types of desmoplastic cancers, which can be used in conjunction with novel anti-CSC or anti-CAF/PSC drugs to improve the treatment efficacy.

## 10. Potential Biomarkers to Guide Next-Generation PanCSC-Targeted Therapies

As highlighted above, anti-CSC therapies should ideally be guided by more pertinent, “stemness-informed” markers of response rather than conventional criteria of treatment response. Changes in such guidelines have already been established in other treatment modalities such as modern immunotherapy in which response rate was rephrased due to pseudo-progression [160,161]. Likewise, stroma- or niche-targeted anti-CSC therapies may also be guided by markers reflecting the pro-stemness properties of tumor stroma. For instance, the density of PSCs in the stroma of PDAC may reflect the abundance level of pro-stemness niches they generated and therefore can serve as a guide for the patient stratification in PSC- or panCSC-targeted therapies. Indeed, α-SMA^+^ CAFs in tumors has been linked to the resistance to neoadjuvant chemotherapy in breast cancer [116]. Activity-specific markers, such as the staining intensity of phosphorylated STAT-1 in CAFs, which our group has shown to reflect the pro-stemness function of CAFs following chemotherapy [38], and the α-SMA^−^PDGF-Rα^+^IL-6^+^ iCAFs in PDAC [115], may also serve as a guide of clinical trials. Conceivably, the clinical utility and predictivity of these stroma-related biomarkers can be further enhanced by combining them with existing markers of panCSCs, such as ALDH, CD133, CD44, CD24, and EpCAM, or the newly identified panCSC regulators such as ASPM, HNF-1A, Msi, and ROR-γ [56,93,94,95,162,163]. We anticipate that the conduction of carefully designed and biomarker-informed clinical trials will maximize the opportunity of successfully developing the next-generation anti-CSC therapy for PDAC.

## 11. Future Directions

The last decade has witnessed the rapid evolution of the CSC model of tumorigenesis, which has shifted away from the static model of CSCs toward a more dynamic and plastic model of cancer stemness. However, our improved understanding of the regulation of CSCs has not been matched by successful clinical developments of CSC-targeted agents. In PDAC, the targeting of panCSCs faces another hurdle of the highly impenetrable desmoplastic stroma, making most therapeutics largely inaccessible to their target tumor cells. These hurdles instigate researchers to adopt novel and alternative strategies to target cancer stemness in PDAC with some of them highlighted in this review. We envisage that the next-generation anti-CSC therapies in PDAC should be designed to target the more accessible stroma cells and niche cells, such as Wnt-producing cells, PSCs, MSCs, and/or TAMs, using highly penetrable small-molecules, nanoparticle-formulated drugs or oligonucleotides, and perhaps immune cell therapy. Alternatively, prior to the clinical approval of anti-CSC therapeutics, some of the pro-stemness functions of stroma cells can be tempered by altering the dosing schedule of systemic chemotherapy to LDM regimens. The stroma- and niche-targeted therapies can be integrated into existing therapies to prevent therapy-induced stromal alterations to improve the outcome of patients with PDAC. Finally, the design of clinical trials at the next stage should be rationally guided by a combination of surrogate markers of PSCs, their activation status, and/or cancer stemness. In addition, the design of new clinical studies should take into consideration combinatorial therapies in order to achieve an acceptable outcome, as targeting solely the CSC or their supporting accessory cells may not be as effective as using a combination therapy when simultaneously targeting tumor and stromal cells. In the next decade, we are about to see whether or not these novel and potentially more viable approaches of CSC targeting may indeed fulfill their promise in the clinic.

## Figures and Tables

**Figure 1 jcm-08-00702-f001:**
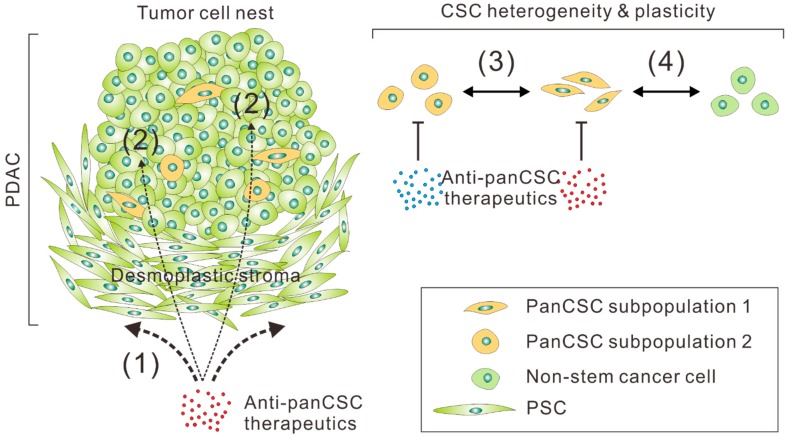
Potential hurdles in pancreatic cancer stem cells (panCSC)-targeted therapy. Due to the desmoplastic reaction of the stroma of pancreatic ductal adenocarcinoma (PDAC) engendered by pancreatic stellate cells (PSCs), which impedes drug penetration, only a small proportion of the anti-panCSC therapeutics can reach their target tumor cells to exert their anticipated effects (1). Even when the therapeutics successfully penetrate the desmoplastic stroma, most of them will reach non-stem cancer cells, which comprise the majority of the cancer cells, rather than the small subpopulation of panCSCs (2). Since panCSCs are highly heterogeneous and comprise partially interconvertible subpopulations, the therapeutics designed to target a specific subpopulation of panCSCs might not be able to inhibit or eradicate other subpopulations of panCSCs (3). Even if all the panCSCs are eradicated by the therapeutics, non-stem cancer cells may be stimulated to transdifferentiate into new pools of panCSCs following the therapy (4), and therefore the tumor regains its cellular heterogeneity and resumes its growth and aggressiveness.

**Figure 2 jcm-08-00702-f002:**
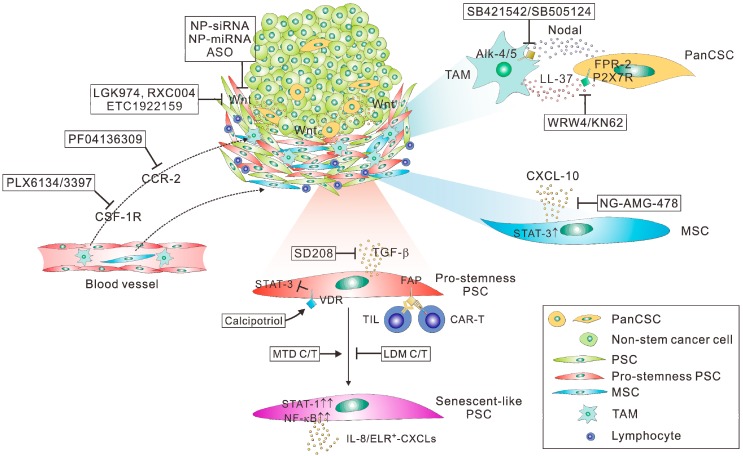
Potentially viable routes to targeting panCSCs and pancreatic cancer stemness. Therapeutics targeting Wnt-related pro-stemness niches, such as the porcupine inhibitors (LGK974, RXC004, and ETC-1922159), nanoparticle (NP)-formulated siRNA or miRNA, and synthetic antisense oligonucleotides (ASO), can prevent activation of Wnt signaling in panCSCs and non-stem cancer cells. Compared with panCSCs, the pro-stemness pancreatic stellate cells (PSCs) or mesenchymal stem cells (MSCs) residing in the tumor stroma or around blood vessels are more accessible to intravenously delivered therapeutics, such as the small-molecule inhibitor of TGF-β (SD208) or VDR signaling (calcipotriol) and novel immunotherapeutic agents, including DNA vaccine of FAP or other PSC-specific antigens, which elicits PSC-specific tumor-infiltrating T cells (TILs), FAP-directed CAR-T cells, and other types of engineered immune cells targeting PSCs. The intra-stromal and intra-tumoral delivery of PSC- or MSC-targeted therapeutics can be enhanced by nanoparticle formulation, such as the nanoghost (NG)-encapsulated CXCL-10 inhibitor AMG487 and nanocarrier-formulated sTRAIL gene therapy. The recruitment of pro-stemness tumor-associated macrophages (TAMs) residing into the PDAC stroma can be inhibited by small-molecules inhibitors of CSF-1R (PLX6134 or PLX3397) or CCR-2 (PF04136309). The TAM–panCSC crosstalk can be inhibited by small-molecular inhibitors of the Nodal/Activin-A receptors Alk-4 and Alk-5 (SB421542 and SB505124) or the inhibitors of the LL-37 receptors FRP-2 (WRW4) and P2X7R (KN62). Finally, low dose metronomic (LDM) chemotherapy can attenuate therapy-induced PSC activation and secretion of pro-stemness chemokines through chronic activation (↑↑) of STAT-1 and NF-κB signaling, including IL-8 and ELR^+^ CXCLs, serving as an immediately clinically deployable strategy to indirectly targeting panCSCs. Note that antibody therapeutics with potential activity in inhibiting panCSCs and/or their niches are not included in the schematic diagram because of their potentially poor stroma penetration and clinical viability in PDAC.

**Table 1 jcm-08-00702-t001:** Clinical-stage agents with potential activity in panCSC targeting described in the current review.

Mode of Action	Example Therapeutics	Clinical Trial Stage	Target Cancer	Clinical Trials.Gov IDs
Porcupine inhibitor	LGK974 (Novartis)	Phase 1 (with PDR001 ^1^)	Wnt-dependent solid tumor	NCT01351103
	RXC004 (Redx Pharma)	Phase 1/2a	Solid tumor	NCT03447470
	ETC-1922159 (A*STAR, Singapore)	Phase 1	Solid tumor	NCT02521844
CSF-1R inhibitor	ARRY382 (Array Biopharma)	Phase 1	Metastatic cancer	NCT01316822
		Phase 1–2 (with pembrolizumab)	Advanced solid tumors	NCT02880371
	PLX3397 (pexidartinib; Plexxikon)	Phase 1 (with durvalumab)	Metastatic/advanced PDAC or CRC	NCT02777710
		Phase 1/2 (with pembrolizumab)	Melanoma and solid tumors	NCT02452424
CCR-2 inhibitor	PF04136309 (Pfizer)	Phase 1 (with FOLFIRINOX chemotherapy ^2^)	Borderline respectable and locally advanced PDAC	NCT01413022
Metronomic chemotherapy	S-1 (Taiho Pharma)	Approved for metastatic PDAC in Asia		

^1^ Anti-programmed death (PD)-1 monoclonal antibody; ^2^ Oxaliplatin, irinotecan, leucovorin, and 5-fluorouracil.

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
