# Peer review of "Next Viable Routes to Targeting Pancreatic Cancer Stemness: Learning from Clinical Setbacks"

_jcm, 2019, doi:10.3390/jcm8050702_

Reviewer 1 Report

The manuscript entitled, “Next viable routes to targeting pancreatic cancer stemness: learning from clinical setbacks” submitted to the Journal of Clinical Medicine is well written and surveys the current clinical landscape in terms of treating pancreatic ductal adenocarcinoma (PDAC). I found the article to be very inclusive and well-focused and feel this is an excellent addition to the field.

Minor edits:

Page 1 : line 28, impeding

Page 9: line 189: suggesting it is CSCs??? (needs rewording)

Figure 2 needs to be enlarged vertically as there is some cramming in the middle

References have minor formatting issues.

Author Response

The manuscript entitled, “Next viable routes to targeting pancreatic cancer stemness: learning from clinical setbacks” submitted to the Journal of Clinical Medicine is well written and surveys the current clinical landscape in terms of treating pancreatic ductal adenocarcinoma (PDAC). I found the article to be very inclusive and well-focused and feel this is an excellent addition to the field.

Minor edits:

Page 1: line 28, impeding

Response 1: We thank the reviewer for pointing out this typo and have corrected it accordingly in the revised manuscript (line 28, page 1).

Page 9: line 189: suggesting it is CSCs??? (needs rewording)

Response 2: We apologize for the phrasing problem in this sentence and have edited it as “a side population of PDAC cells has been shown to resist gemcitabine therapy and express stemness-associated and prognostic genes” in the revised manuscript (line 188-189, page. 9).

Figure 2 needs to be enlarged vertically as there is some cramming in the middle

Response 3: We agree with the reviewer that the middle part of Figure 2 is a bit cramming and have thus expanded it to make the different parts of the figure more discrete in the revised Figure 2.

References have minor formatting issues.

Response 4: We have re-formatted the references according to the MDPI reference style template using the Endnote software.

Reviewer 2 Report

I think the manuscript is good. It is a comprehensive review and well written. The review paper covers many of the key topics in field of pancreatic CSC. I think it may be better if the paper can add a section to discuss targeting the crosstalk between pancreatic CSC and pancreatic CAF or pancreatic stellate cells.

Author Response

I think the manuscript is good. It is a comprehensive review and well written. The review paper covers many of the key topics in field of pancreatic CSC. I think it may be better if the paper can add a section to discuss targeting the crosstalk between pancreatic CSC and pancreatic CAF or pancreatic stellate cells.

Response 1: In our original manuscript, we have included a whole section (6. Route 2: Targeting pro-stemness PSCs) describing the role of pancreatic stellate cells (PSCs), a specialized type of CAFs in pancreatic cancer, in the maintenance of panCSCs and their molecular cross-talks (starting from line 445, page 20). We then summarized several small-molecule inhibitors that may inhibit PSCs or their communication with panCSCs, including the TGF-b inhibitor SD208, the vitamin D receptor agonist, calcipotriol, and novel approaches to deplete PSCs, including DNA vaccine and CAR-T cells. To satisfy the reviewer, we further added a new paragraph summarizing other therapeutics, including anti-IL-6 and anti-LIF antibodies, that can potentially block the PSC-panCSC crosstalk, as well as the inhibitors of CXCR-2 or STAT-3, that may direct inhibit the downstream signaling events in panCSCs in the revised manuscript (lines 483-494, page 22). We also bring to the attention of the readers that large-molecules like antibodies have very limited penetration in desmoplastic cancers like pancreatic cancer, and that the direct targeting of panCSCs is challenging due to their highly dynamic and plastic characteristics (lines 494-501; page 22). These potential hurdles highlight the need of developing novel stemness-targeted therapeutics as summarized in our manuscript. Therefore, we would like to respectfully bring to the reviewer’s attention that the main goal of this review is to provide a balanced view of the assorted drugs currently in developments and to suggest potentially viable routes of targeting panCSCs and/or their niches that may have a higher success rate in clinical trials.

J. Clin. Med. EISSN 2077-0383 Published by MDPI AG, Basel, Switzerland RSS E-Mail Table of Contents Alert
Back to Top